# A Metabolomic Approach to Unexplained Syncope

**DOI:** 10.3390/biomedicines12112641

**Published:** 2024-11-19

**Authors:** Susanna Longo, Ilaria Cicalini, Damiana Pieragostino, Vincenzo De Laurenzi, Jacopo M. Legramante, Rossella Menghini, Stefano Rizza, Massimo Federici

**Affiliations:** 1Department of Systems Medicine, University of Rome Tor Vergata, Via Montpellier 1, 00133 Rome, Italy; susanna.longo@uniroma2.it (S.L.); legraman@uniroma2.it (J.M.L.); menghini@uniroma2.it (R.M.); rizza@med.uniroma2.it (S.R.); 2Department of Innovative Technologies in Medicine and Dentistry, “G. d‘Annunzio” University of Chieti-Pescara, 66100 Chieti, Italy; ilaria.cicalini@unich.it (I.C.); damiana.pieragostino@unich.it (D.P.); vincenzo.delaurenzi@unich.it (V.D.L.); 3Center for Advanced Studies and Technology (CAST), “G. d‘Annunzio” University of Chieti-Pescara, 66100 Chieti, Italy

**Keywords:** unexplained syncope, metabolomics, cardiometabolic risk factors, cardiovascular diseases, glutamine, lysine, lysophosphatidylcholine

## Abstract

**Background:** This study aims to identify a metabolomic signature that facilitates the classification of syncope and the categorization of the unexplained syncope (US) to aid in its management. **Methods:** We compared a control group (CTRL, *n* = 10) with a transient loss of consciousness (TLC) group divided into the OH group (*n* = 23) for orthostatic syncope, the NMS group (*n* = 26) for neuromediated syncope, the CS group (*n* = 9) for cardiological syncope, and the US group (*n* = 27) for US defined as syncope without a precise categorization after first- and second-level diagnostic approaches. **Results:** The CTRL and the TLC groups significantly differed in metabolic profile. A new logistic regression model has been developed to predict how the US will be clustered. Using differences in lysophosphatidylcholine with 22 carbon atom (C22:0-LPC) levels, 96% of the US belongs to the NMS and 4% to the CS subgroup. Differences in glutamine and lysine (GLN/LYS) levels clustered 95% of the US in the NMS and 5% in the CS subgroup. **Conclusions:** We hypothesize a possible role of C22:0 LPC and GLN/LYS in re-classifying US and differentiating it from cardiological syncope.

## 1. Introduction

Syncope is a transient loss of consciousness (TLC) characterized by rapid onset, short duration, and spontaneous reversibility with complete recovery. It is a symptomatic consequence of global cerebral hypoperfusion triggered by vasovagal stimulation, arterial hypotension, or low systolic output. Despite its high frequency, obtaining an accurate estimate of the incidence of syncope is difficult because of the different definitions used among studies and because most syncopal events do not require medical attention [1]. According to the guidelines of the European Society of Cardiology (ESC) of 2018 [2], syncope is classified as follows:

reflex or neurologically mediated syncope (NMS), related to a specific trigger;syncope due to orthostatic hypotension (OH), defined as a drop > 20 mmHg in systolic blood pressure (SBP) or >10 mmHg in diastolic blood pressure after standing for three minutes;cardiac syncope (CS), caused by arrhythmic pathologies or structural diseases of the heart and great vessels.

NMS can be further classified into vasovagal syncope (emotional or due to prolonged standing), situational syncope (related to a specific trigger), syncope due to stimulation of the carotid sinus (carotid sinus syndrome), and non-classical forms that occur without the typical prodromes or triggers of NMS [2]. To clarify the pathophysiological mechanisms underlying these events and to re-classify some unexplained syncopal events as NMS, the evaluation of plasma adenosine levels and the characterization of the adenosine A2A receptor have been proposed [3]. Specific purinergic profiles characterize different forms of NMS that can be classified as low-, normal-, and high-adenosine syncope. This classification not only provides a deeper understanding of NMS but also offers hopeful therapeutic implications.

Syncopal episodes that are difficult to categorize according to the guidelines or the metabolic classifications are often found even after further diagnostic investigations in the Syncope Units. These episodes remain as unexplained syncope (US) [4]. US represents a crucial challenge in identifying the risk of short-term adverse outcomes, as it is difficult to determine whether the event will follow a benign course, such as in NMS, or if it is at risk of major cardiovascular events.

The occurrence of syncope throughout a person’s lifetime is estimated to be at least 32–35% [5,6,7], with a higher incidence in women and the elderly [8]. Its incidence in the Framingham Heart Study (1971–1998) was 6.2 per 1000 person-years, and the most frequent etiology was NMS (21.2% of cases), followed by CS and OH (9.5% and 9.4%, respectively). The incidence of US was over one-third of cases (36.6%) [9]. The syncope Emergency Department (ED) admission rate is about 0.9–1.7% [10,11]. Admission rates from the ED to the hospital vary across countries and health systems (ranging from 12 to 15% in Canada [12,13], from 31 to 38% in Italy [14,15], from 49% in the United Kingdom [16], and 46–86% in the United States of America [17,18]) and result in substantial healthcare costs [14,15,19]. Thus, the economic impact of syncope treatment can be very high [20].

One of the main difficulties in managing syncope is the wide range of etiologies that can cause it [21].

According to the main guidelines [21,22,23], the first approach to a patient suffering from syncope requires the collection of an accurate medical history and the execution of a careful physical examination with orthostatic blood pressure measurements, an electrocardiogram (ECG), and additional tests when needed [24]. After this preliminary evaluation, syncope can be distinguished from other forms of TLC [2], and the etiological diagnosis can be defined in about half of the cases [25].

When syncope is unexplained at initial evaluation in the ED, further investigation is necessary based on risk stratification for short-term adverse events. The ESC guidelines [2] suggest evaluating the presence of a history of severe structural, coronary heart disease, significant comorbidities, clinical features, and arrhythmias on the ECG as high-risk criteria for short-term adverse events, which require prompt hospitalization or intensive evaluation in a hospital setting. Otherwise, patients are considered at low risk and can continue investigations in outpatient pathways if deemed by the clinician [2]. The Canadian Cardiovascular Society (CCS) Position Paper [22] and the American College of Emergency Physicians (ACEP) Clinical Policy [23] stratify short-term adverse event risk based on cardiovascular risk factors, age, and associated comorbidities [24].

The scenario is much more complicated in clinical practice. The incidence of US is about 37–40%, and, despite thorough evaluation, up to 60% of patients will not have an official diagnosis at initial evaluation in the ED [21]. Furthermore, there is no optimal approach to risk stratification because the tools proposed to reduce unnecessary hospitalizations and syncope-related healthcare costs have significant limitations [25]. Indeed, according to the ESC guidelines [2], currently available risk stratification scores should not be used alone to predict severe short-term outcomes after syncope because they lack more diagnostic accuracy or prognostic performance than clinical judgment.

Therefore, categorizing US can be difficult. Furthermore, US is considered an episode of CS until proven otherwise and individuals with US undergo expensive and invasive evaluations that may not lead to an official diagnosis. A better understanding of US can help in the evaluation process by avoiding unnecessary and expensive tests [26,27,28,29,30,31,32,33].

In our study, we applied an omics technique to deepen our knowledge of the true nature of US and suggest a practical application in hospital management. This study aims to identify a metabolomic signature that facilitates the classification of syncope and the categorization of US to aid in its management.

## 2. Materials and Methods

### 2.1. Patients

Our cross-sectional pilot study evaluated 104 patients attending the Day Hospital Service at Tor Vergata University Hospital (Rome, Italy) from February 2020 to May 2022. The inclusion criteria were the presence of non-high-risk syncope according to the ESC 2018 [2]; age between 25 and 89 years old; body mass index (BMI) > 18; good compliance with the research protocol; and ability to independently understand and sign the informed consent. The exclusion criteria were the presence of pregnancy or breastfeeding; major psychiatric disorders; stage IV renal failure [indicated by estimated glomerular filtration rate (eGFR) < 30 mL/min]; liver disease or liver failure (indicated by abnormal values of parameters such as alanine aminotransferase, aspartate aminotransferase, gamma-glutamyl transferase, alkaline phosphatase or blood bilirubin > 5 times the upper reference value); donation of blood or blood products (i.e., >450 mL of plasma or platelets) immediately before the start of the study and after 4–12 weeks; participation in other protocol during the four weeks before administration of informed consent; cognitive assessment by Mini-Mental State Examination (MMSE) < 24 corrected for age.

### 2.2. Clinical, Metabolic Parameters Assessment, and Instrumental Exams

A complete clinical history was recorded regarding lifestyle habits and comorbidities (i.e., diabetes, cardiovascular disease, arterial hypertension). Anyone who regularly smoked at least one cigarette per day was counted as a current smoker, and current smokers and former smokers were counted together as a single group compared to non-smokers. Alcohol consumption was recorded as the number of drinks per day. Participants who engaged in physical activity for at least 1 h per week were considered physically active. The following clinical and metabolic anthropometric variables were measured: height, weight, waist circumference, random blood pressure (BP), and BMI calculated by dividing weight (in kilograms) by the square of height (in meters). BP was measured in the dominant arm in a seated position with a standard sphygmomanometer cuff of an appropriate size.

Carotid intima-media thickness (c-IMT) was calculated using the Esaote Mylab system (Ref 101620000, Esaote SPA, Genoa, Italy) with a VF 13 × 10^−5^ linear array transducer. Anterior, lateral, and posterolateral views were used to visualize the right and left common carotid arteries longitudinally. At each projection, three determinations of c-IMT were carried out at 2 cm proximal to the bulb, in the site of the most significant thickness. The values at each site were averaged.

Approximately 30 mL of whole blood was drawn between 8:00 and 9:00 a.m. after an overnight fast. Blood samples were collected after an overnight fast to prevent test result interference. It was used to perform routine laboratory evaluations, including complete blood count, total cholesterol, HDL cholesterol, triglycerides, serum creatinine, eGFR, and glycemia.

### 2.3. Metabolomic Assessment

About 8 mL of whole blood underwent centrifugation for serum sampling for 40 min at 4 °C at 1000 rcf. Each sample was immediately packaged in a 5.0 mL Eppendorf Tubes^®^ (Eppendorf S.r.l., Milan, Italy) container and stored at −80 °C until the evaluation of metabolites. Then, 125 μL of extraction solutions containing an internal standard provided by the NeoBase 2 Non-Derivatized MSMS Kit (Perkin Elmer Life and Analytical Sciences, Turku, Finland) were added to 10 µL of plasma samples for the detection of amino acids, acylcarnitines (ACs), free carnitines, succinylacetones, nucleosides, and lysophospholipids. After 30 min of incubation at 45 °C at 700 rpm, 100 μL of the solution was transferred onto a transparent plate. The MS/MS system consists of RenataDX™ Screening Systems (Waters Corporation, Milford, MA, USA) as a fully FIA-MS/MS IVD system for high-throughput analysis. A 3777C IVD Sample Manager, ACQUITY™ UPLC™ I-Class IVD Binary Solvent Manager (Waters Spa Sesto San Giovanni, Milan, Italy), and Xevo™ TQD IVD Mass Spectrometer (Waters Corporation, Milford, MA, USA) were used online for FIA-MS/MS analysis. A total of 10 μL were injected into the ion source, and the run time was 1.3 min, injection-to-injection, as already described [34,35]. Mass spectra were processed using MassLynx ™ (IVD) Software V4.2 with IonLynx ™ Application Manager (Waters Corp, Milford, MA, USA). The complete list of all analyzed metabolites, with their respective abbreviations and mass transitions used for quantification, is described in Appendix A.

### 2.4. Statistical Analysis

Statistical analysis for clinical data was performed with Jamovi version 2.3.21 [36,37]. Continuous variables were measured as mean or median ± standard deviation, according to data distribution; categorical data were expressed as a percentage of frequency. The normal distribution was assessed with the Shapiro–Wilk test. Analysis of Variance (one-way ANOVA) or the Kruskal–Wallis test for quantitative variables and χ^2^-test for categorical variables were used to test the significance of between-group comparisons. For all these analyses, a *p*-value < 0.05 was considered statistically significant.

Regarding the metabolomic analysis, a Partial Least-Squares Discriminant Analysis (PLS-DA) and heatmaps were performed to compare the CTRL and TLC groups using the Metaboanalyst 5.0 free-on-line tool. Pearson correlation was performed to evaluate the correlation between the plasma metabolic profile and clinical characteristics which differed significantly between the five groups. Unpaired Student’s *t*-test and Analysis of Variance (one-way ANOVA) were performed to identify individual modulated metabolites with subsequent post hoc Tukey and false discovery rate (FDR) tests. An FDR < 0.05 was considered statistically significant. ROC curve-based model evaluation (Tester) tools (Metaboanalyst 5.0) were used for biomarker analysis with internal validation based on 100-cross validation and new sample class prediction. GraphPad Prism 7.0 was used for outlier calculations and for creating histograms, box plots, and dot plots.

## 3. Results

According to the selection criteria, we screened 105 subjects with syncope; 104 were included, and the informed consent was signed. A total of 19 patients were excluded due to poor adherence to the study protocol. Consequently, 85 participants completed the study protocol and were enclosed in the TLC group. They were divided into three subgroups according to the 2018 ESC classification of the pathophysiology of syncope [2]: the OH group (*n* = 23) for orthostatic syncope, the NMS group (*n* = 26) for neuromediated syncope, and the CS group (*n* = 9) for cardiological syncope. A fourth subgroup was identified, including subjects with syncope whose presentation did not fall into any ESC guideline category and without a precise diagnosis after first- and second-level diagnostic approaches: the US group (*n* = 27) [2]. Their data were compared with a group of healthy subjects screened in the same Day Hospital Service and who had not experienced any syncope, named the CTRL group (*n* = 10).

### 3.1. Clinical Profile

Table 1 summarizes the five groups’ anthropometric, clinical, and metabolic characteristics. All five groups did not differ significantly for anthropometric data (age, sex, weight, height, BMI, waist, and diastolic BP) and metabolic parameters (glucose, eGRF, lipid profile, hemoglobin, and white blood cell count). Furthermore, the groups were comparable in non-modifiable risk factors (history of diabetes, hypertension, and cardiovascular disease) and modifiable risk factors (alcohol consumption and exercise), except for smoking (*p*-value 0.017). They also differed in SBP measured at the first visit (*p*-value 0.005) and c-IMT (*p*-value 0.001). Considering this evidence, we performed a Pearson correlation between plasma metabolic data and SBP and c-IMT values, observing no moderate or strong correlation [38], as evidenced by the Pearson coefficients reported in Table 2.

### 3.2. Metabolomic Profile

We explored the metabolomic profile of the five groups; the values of the measured metabolites as well as the values of the metabolic ratios are reported in Appendix A.

During this stage, one subject was excluded due to high Phenylalanine (PHE) values. After performing a PLS-DA, we identified modulated metabolites by directly comparing the CTRL and TLC groups’ metabolomic profiles. These difference concerned Ornithine (ORN, *p*-value 0.002), Valine (Val, *p*-value 0.002), Leucine-Isoleucine and Hydroxyproline (LEU-ILE-PRO-OH, *p*-value 0.002), Proline (PRO, *p*-value 0.013), Methionine/Phenylalanine ratio (MET/PHE, *p*-value 0.014), acylcarnitine C18 (*p*-value 0.02), C18:2 (*p*-value 0.02), Leucine/alanine ratio (LEU/ALA, *p*-value 0.025), glutamate (GLU, *p*-value 0.028), free carnitine/acylcarnitines ratio (C0/(C16 + C18), *p*-value 0.028), acylcarnitine C18:1 (*p*-value 0.032), Arginine (ARG, *p*-value 0.033), acylcarnitine C5:1 (*p*-value 0.035), and Methionine (MET, *p*-value 0.042) [Appendix A], but they lost statistical significance by calculating the false discovery rate (FDR), as reported in the Appendix A.

Statistically significant differences also emerged when comparing each TLC subgroup with the CTRL group. The metabolic profile of CS differed from that of the CTRL group for PHE (*p*-value 3.42 × 10^−4^; FDR = 0.009), Phenylalanine/Tyrosine ratio (PHE/TYR) (*p*-value 2.48 × 10^−4^; FDR = 0.009), ARG (*p*-value 9.34 × 10^−4^; FDR = 0.015), Citrulline (CIT) (*p*-value 0.002; FDR = 0.026), ORN (*p*-value 7.9 × 10^−4^; FDR = 0.015), C18 (*p*-value 2.05 × 10^−4^; FDR = 0.009), and acylcarnitine C18OH (*p*-value 0.001; FDR = 0.018). The cumulative ROC indicated an AUC = 0.948 [Figure 1A–C, Appendix A]. The metabolic profile of OH differed from that of the CTRL group for PHE (*p*-value 2.02 × 10^−4^; FDR = 0.017) and for PRO, which loses statistical significance after calculating FDR (*p*-value 0.02; FDR = 0.10). The cumulative ROC indicated an AUC = 0.888 [Figure 1D–F, Appendix A]. The metabolic profile of the US group differed from that of the CTRL group for PHE (*p*-value 0.001; FDR = 0.047) and PHE/TYR (*p*-value 9.01 × 10^−4^; FDR = 0.047). The cumulative ROC indicated an AUC = 0.916 [Figure 1G–I, Appendix A]. Considering the comparison between the NMS group and the CTRL, a significant modulation was observed for LEU-ILE-PRO-OH and VAL (*p*-values 0.001 and 0.002, respectively). After calculating FDR, no differences were found between the CTRL and the NMS group, and the cumulative ROC indicated an AUC = 0.793 [Figure 1J–L, Appendix A].

Thus, we compared the metabolomic profiles of the four TLC subgroups directly with each other. PLS-DA analysis indicated homogeneous metabolomic profiles among the TLC subgroups, suggesting no confounding factors [Figure 2A]. The heatmaps confirmed this evidence. Focusing on the US group, it could not be grouped into a defined class but was homogeneously dispersed within the CS, NMS, and OH clusters in a direct comparison [Figure 2B–D].

After analyzing the metabolite levels between different subgroups, we developed a new logistic regression model to predict how to cluster US.

From the comparison between OH and NMS, both acylcarnitine with 24 carbon atoms (C24) and lysophosphatidylcholine with 22 carbon atoms (C22:0-LPC) were increased in NMS when compared to OH (*p*-value 0.048 and *p*-value 0.040, respectively) [Figure 3A,B], but with a poor diagnostic accuracy at cumulative ROC (AUC 0.631) [Figure 3C]. Applying the logistic regression model with C24 and C22:0-LPC, as reported in Appendix A, 52% of subjects with US clustered in the NMS group and 48% in the OH group [Figure 3D].

Comparing the OH and CS groups, several acylcarnitines were statistically significantly increased in the CS group (C18OH *p*-value 0.01, C10:2 *p*-value 0.02, C12:1 *p*-value 0.03, C14OH *p*-value 0.03), as reported in Appendix A, but all together, they demonstrated poor diagnostic ability in distinguishing the two subgroups (AUC = 0.573).

In the same comparison, argininosuccinic acid (ASA) and ARG increased in the CS subgroup (*p*-value 0.02; *p*-value 0.04, respectively) [Figure 4A,B]. As shown in Appendix A, applying the logistic regression model with ASA and ARG plasma levels pooled together, 78% of the US group were clustered in the OH group, while 22% were in the CS group [Figure 4C].

From the comparison between the CS and the NMS groups, the long-chain acylcarnitines C18OH, C18, C14OH, C10, C4, and C14 (respectively, *p*-value 0.001; *p*-value 0.02; *p*-value 0.02; *p*-value 0.03; *p*-value 0.03; and *p*-value 0.03) were significantly increased in the CS group. In the same comparison, acylcarnitines C24, C12:1, and C12 were significantly increased in the NMS group (respectively, *p*-value 0.03; *p*-value 0.04; and *p*-value 0.04), as reported in Appendix A. Upon applying the logistic regression model, the model lacked the necessary robustness to be generated. Furthermore, the long-chain acylcarnitines pooled together did not exhibit the necessary diagnostic ability to distinguish between the two subgroups, with an AUC of 0.61.

In the comparison between the CS and the NMS groups, the summation of glutamine and lysine (GLN/LYS) was increased in the CS subgroup (*p*-value 0.01), and C22:0-LPC was increased in the NMS subgroup (*p*-value 0.03) [Figure 5A,B]. Mass spectrometry analysis cannot distinguish between GLN and LYS because they have the same mass/charge ratio; they are considered a summary. Applying the logistic regression model, GLN/LYS clustered 95% of the US group in the NMS group and 5% in the CS group [Figure 5C]; the C22:0-LPC model clustered 96% of the US group in the NMS group and 4% in the CS group [Figure 5D]. Appendix A details the logistic regression algorithm for the GLN/LYS and C22:0-LPC models, respectively.

## 4. Discussion

The metabolomic approach demonstrated excellent diagnostic and prognostic accuracy in differentiating subjects with syncope from the CTRL group and in the re-classification of the US group. The new sample prediction logistic regression model built to predict how to cluster US based on the differences in C22:0-LPC levels allows us to re-classify 96% of the US patients in the NMS subgroup and 4% in the CS subgroup, while GLN/LYS clustered 95% of the US subjects in the NMS group and 5% in the CS subgroup.

LPCs are a class of lipid biomolecules derived both from the hydrolysis of phosphatidylcholine (PC) in tissues by the enzyme phospholipase A2 (PLA2) and from the transfer of fatty acids to free cholesterol in the plasma by the action of lecithin-cholesterol acyltransferase (LCAT). LPCs can be reconverted by the enzyme lysophosphatidylcholine acyltransferase (LPCAT) in the presence of Acyl-CoA into PC, which is essential for forming biological membranes [39]. LPC receptors belong to the G protein-coupled and Toll-like receptors family [40].

The role of LPCs is twofold: they perform many biological functions, such as regulating the expression of critical factors in the production of lipoproteins, but they also play a crucial role in the development of several diseases, such as atherosclerosis and inflammatory diseases [21]. LPCs perform a proinflammatory function by binding to specific receptors, inducing cell division, and releasing inflammatory factors and oxidative stress [40]. Thus, LPCs are linked to the development of pathologies such as Alzheimer’s disease, neuropathic pain, and adrenoleukodystrophy [41].

Although LPCs are involved in several pathological scenarios, they are increasingly recognized as a critical factor associated with neurodegenerative and cardiometabolic diseases due to their proinflammatory and pro-atherogenic actions [39]. While high levels of some LPCs could be associated with diabetes and its complications, such as diabetic retinopathy and neuropathy [41], the reduction in levels of some LPCs in response to an oral glucose tolerance test (OGTT) correlated with the incidence of cardiovascular diseases (CVD) after a 25-year follow-up in a subsample of Framingham Heart Study participants free from diabetes [42]. Moreover, studies confirm that LPCs were involved in insulin resistance induced by saturated fatty acids [43].

In vivo, LPCs promote atherosclerosis by affecting vascular endothelial cells through the mobilization of intracellular calcium, overexpression of growth factors and adhesive molecules, and release of inflammatory factors, increasing oxidative stress and promoting apoptosis [41]. Modified low-density lipoprotein (LDL), enzymatically degraded LDL, and oxidized LDL have a higher content of LPC [39]. Furthermore, LPC levels were significantly higher in atherosclerotic plaques of symptomatic subjects and correlated with markers of oxidative stress and proinflammatory cytokines, which influence the instability of the plaque [44,45]. LPC regulates vascular tone [46,47] and higher levels of LPCs are associated with prehypertension, while lower levels are linked to atherosclerosis and aortic dissection [48,49].

In the KORA study, higher serum LPC levels were linked to a lower risk of myocardial infarction and combined CVD events [50]. Ganna et al. found that LPCs were also associated with lower BMI and C-reactive protein, indicating a protective effect on cardiovascular risk [51]. Furthermore, LPCs produced by the PLA2-like activity of paraoxanase 1 (PON1) in atherosclerotic lesions inhibited macrophage cholesterol accumulation and atherogenesis [52]. In the cardiovascular cohort of the Malmo Diet and Cancer study, higher LPC levels were associated with a reduced CVD risk over a 12-year period and were negatively correlated with carotid IMT, glycosylated hemoglobin, BMI, and SBP and positively correlated with high density lipoprotein (HDL) cholesterol [53]. Stegemann et al. also demonstrated an inverse association between different LPC species and the incidence of coronary heart disease [54]. Given the pro-atherogenic activity of LPCs and their high content in plaque, these results would seem counterintuitive. However, this may reflect the increased catabolism and clearance of LPCs from the circulation as modified lipoproteins or directly from albumin, representing the primary form of plasma LPCs [55]. Another explanation could be the alteration in the synthesis of LPCs, as proposed by some studies showing an association between LCAT deficiency and accelerated atherogenesis. Homozygotes for deleterious mutations in the LCAT gene are characterized by an almost complete deficiency of HDL cholesterol, while heterozygotes have reduced HDL cholesterol levels compared to those expected. Furthermore, carriers of LCAT gene mutations have increased c-IMT, as assessed by magnetic resonance imaging [56]. The key to explaining these controversial results is probably to be found in the complexity of the enzymatic cascade involved in the metabolism of LPCs and in the imbalance that can disrupt the homeostasis of LPCs, leading to metabolic disorders [39].

Conflicting evidence exists regarding the role of LYS and GLN in CVD. LYS is an essential amino acid introduced with the diet and minimally produced by the gut microbiota [57,58]. It is necessary for the organism’s growth and for making a positive nitrogen balance [57]. GLN is synthesized from glutamic acid and ammonia by glutamine synthetase, and it is the most abundant amino acid in the body, so it is conditionally essential during catabolic stress and critical illness [59]. GLN plays a fundamental role in nitrogen exchange between organs and is involved in the biosynthesis of various substances. Additionally, it has significant immunomodulatory effects, contributing to lymphocyte proliferation and maintaining intestinal membrane integrity [59,60].

GLN is also involved in stabilizing atherosclerotic plaques, promoting the polarization of M2 macrophages, and synthesizing glutathione [53]. Low levels of GLN can promote toxic effects on cells by excess reactive oxygen species (ROS) [61]. Low levels of GLN and LYS are associated with increased cardiovascular risk and higher levels of central SBP and arterial stiffness [61,62]. Furthermore, balancing GLN and glutamate is important for energy production in cells, with implications for myocardial metabolism and cardiomyocyte contractility [60,63].

As already postulated in our previous study [64,65], the role of the intestinal microbiota in mediating the association between microbial metabolites and CVD cannot be excluded. Phenylacetylglutamine (PAG), formed by the bacterial conjugation of GLN with phenylacetic acid (derived from dietary Phenylalanine by the gut microbiota), seems to be associated with CVD and the incident risk of major adverse cardiac events such as heart attack, stroke, and death independently from traditional CVD risk factors in both diabetics and non-diabetics. The action of PAG on cardiovascular events is thought to be attributable to its ability to interact with G protein-coupled receptors, including α- and β-adrenergic receptors [66,67].

Based on our evidence, we suggest using metabolomics to classify syncope and categorize US to aid in its management. Metabolomics is a crucial component of precision medicine for adapting and personalizing medical treatment. It uses mass spectrometry and nuclear magnetic resonance to measure metabolites in cells, tissues, or biological fluids [68,69]. This approach can identify the metabolic fingerprints of individuals and specific groups. Metabolomics has been used to determine the metabolic phenotype of various pathologies and is increasingly used to develop new risk stratification approaches in CVD [70,71]. In addition, the method we presented is already in clinical use for the evaluation of inherited metabolic diseases in the context of the expanded newborn screening program, as reported in several papers [72,73,74]. The analysis times are very fast, considering that metabolic profile analysis is performed in less than two minutes, allowing us to provide the patient’s metabolic profile in less than two hours, considering sample preparation and analysis steps. This represents a huge strength, considering its application in the clinical field to assist physicians in a rapid and precise diagnosis, especially in the perspective of precision medicine.

## 5. Limitations

A significant limitation of our study is the small sample size. This study was conducted during the first and second SARS-CoV2 pandemic, severely limiting access to normal hospital pathways for non-infected patients. The same limitations have made it difficult to collect healthy patients to include in the CTRL groups and subjects with CS. Patients were enrolled in the Day Hospital Service, where patients with suspected CS but at low risk of short-term adverse outcomes were referred. Most patients with CS are at high risk of short-term adverse outcomes and are therefore admitted to a cardiology care setting. However, we included the CS group in our analysis to make our research more comprehensive. We believe that the findings of this pilot study will pave the way for further research with a larger sample, allowing us to validate the model used and potentially make significant contributions to the field.

Metabolomics faces limitations due to the lack of standardized techniques, methods, and data reporting across studies, as well as the heterogeneous nature of the biological samples [70]. Standardization of sample collection, extraction, and processing methods is needed for reproducibility. Factors such as age, sex, diet, and medication strongly influence sample collection for metabolomic analysis. Additionally, integrating different data levels and building comprehensive medical databases are essential for applying metabolomics in real-time analysis by machine learning and AI systems [71].

Our findings strongly indicate that C22:0-LPC and GLN/LYS are highly effective tools for re-classifying US. Nonetheless, due to the limitations highlighted earlier, further evaluations are essential to fully confirm our pilot study’s results.

The need to find new biomarkers opens the way to new diagnostic strategies not conventionally used in the study of syncope. Another aspect to consider is how oxidative stress influences the pathophysiological events that trigger syncope through subclinical inflammation, endothelial dysfunction, and autonomic dysfunction. Additionally, it is important to explore whether oxidative stress biomarkers can be a valid aid in the diagnostic workup of US [75].

## 6. Conclusions

In managing syncope, patients without a precise diagnosis after a first and second-level diagnostic approach need new risk stratification strategies and diagnostic tools to avoid excessive or insufficient preventive therapies. A metabolomic approach to syncope management could detect cardiovascular and metabolic biomarkers for application in clinical settings. Based on our results, we hypothesize a possible role for C22:0 LPC and GLN/LYS in re-classifying US and differentiating it from CS.

## Figures and Tables

**Figure 1 biomedicines-12-02641-f001:**
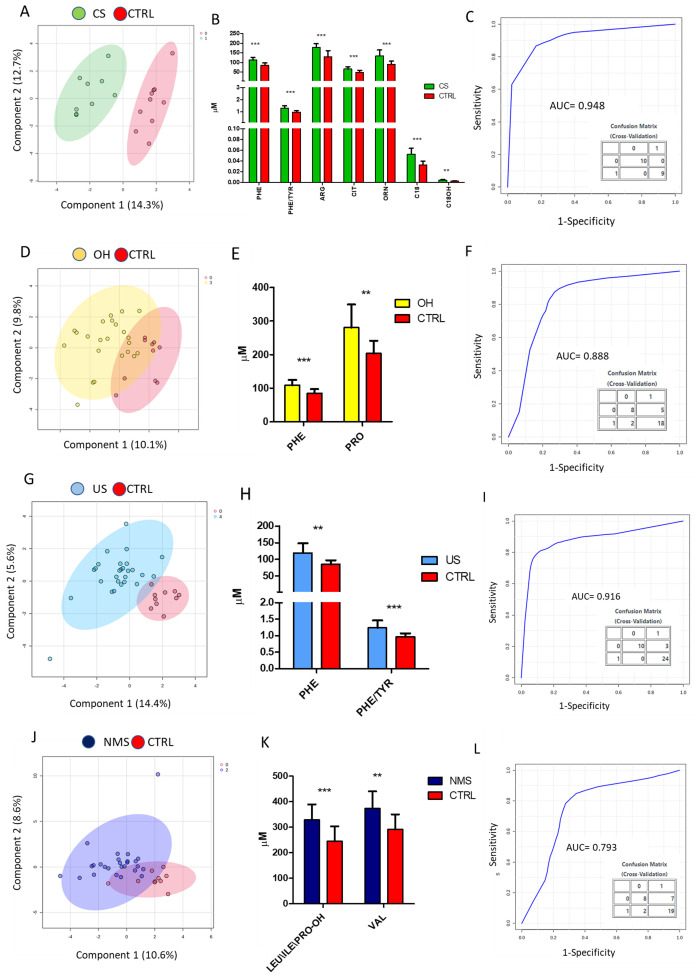
PLDS-DA analysis between the CTRL and each TLC group. PLDS-DA analysis based on the metabolites pattern quantified in the CTRL and CS groups (**A**); the plasma levels of Phe, Phe/Tyr, Arg, Cit, Orn, C18, and C18OH quantified in the comparison between the CS and CTRL groups (**B**); cumulative ROC curve based on the significantly modulated metabolites depicted in (**B**) (CS vs. CTRL) and the associated Confusion Matrix resulting from the predicted class probability across the 100 cross-validation (**C**); PLDS-DA analysis based on the metabolites pattern quantified in the CTRL and OH groups (**D**); the serum levels of Phe and Pro quantified in the comparison between the OH and CTRL groups (**E**); cumulative ROC curve based on the two significantly modulated metabolites depicted in panel E (OH vs. CTRL) and the associated Confusion Matrix resulting from the predicted class probability across the 100 cross-validation (**F**); PLDS-DA analysis based on the metabolites pattern quantified in the CTRL and US groups (**G**); the serum levels of Phe and Phe/Tyr in the comparison between the US and CTRL groups (**H**) and the related cumulative ROC and associated Confusion Matrix resulting from the predicted class probability across the 100 cross-validation (**I**); PLDS-DA analysis based on the metabolites pattern quantified in the CTRL and NMS groups (**J**); the serum levels of Leu-Ile-Pro-OH and Val in the comparison between the NMS and CTRL groups (**K**) and the related cumulative ROC and associated Confusion Matrix resulting from the predicted class probability across the 100 cross-validation (**L**). ** *p*-value < 0.01, *** *p*-value < 0.001.

**Figure 2 biomedicines-12-02641-f002:**
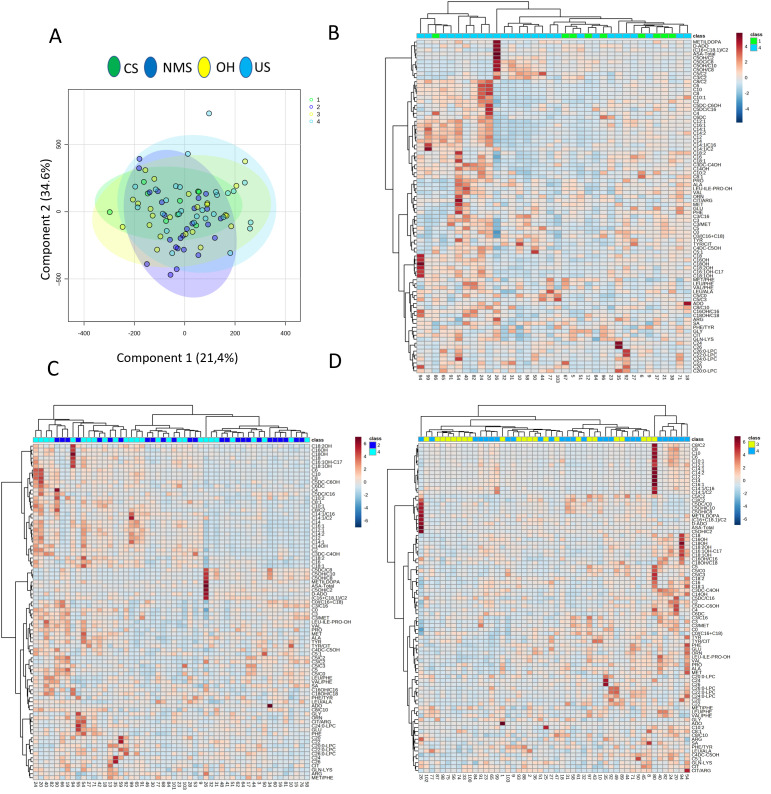
PLDS-DA analysis and heatmaps based on the comparison between each TLC group. PLDS-DA analysis based on the metabolite pattern quantified in the TLC groups, in particular, CS in green, NMS in dark blue, OH in yellow, and US in light blue (**A**); heatmaps based on serum metabolite levels quantified in CS (**B**), NMS (**C**), and OH (**D**) for evaluation of the US group cluster analysis.

**Figure 3 biomedicines-12-02641-f003:**
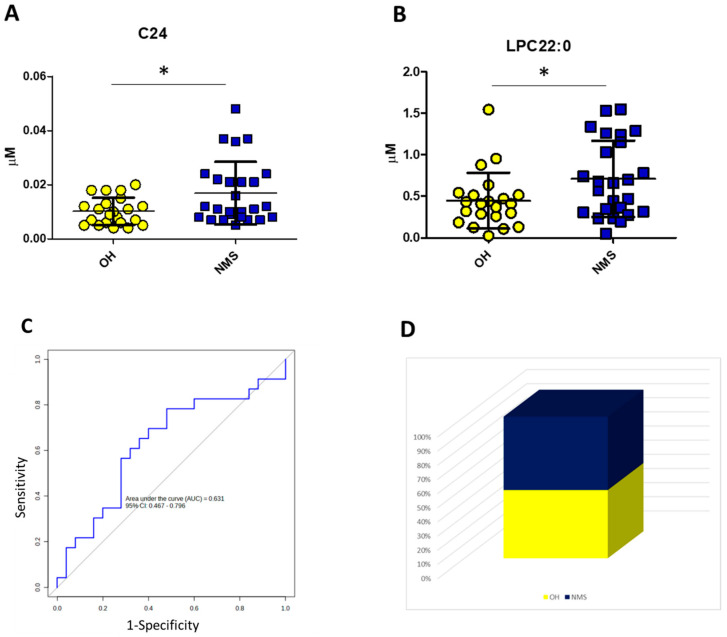
The logistic regression model with C24 and C22:0-LPC. Dot-plot showing plasma levels of C24 (**A**) and C22:0-LPC (**B**) in the comparison between the OH and NMS groups; cumulative ROC built by using C24 and C22:0-LPC simultaneously based on a logistic regression model for the comparison between OH and NMS (**C**); re-classification of US patients based on the application of the logistic regression algorithm described in Table 2 (**D**). * *p*-value < 0.05.

**Figure 4 biomedicines-12-02641-f004:**
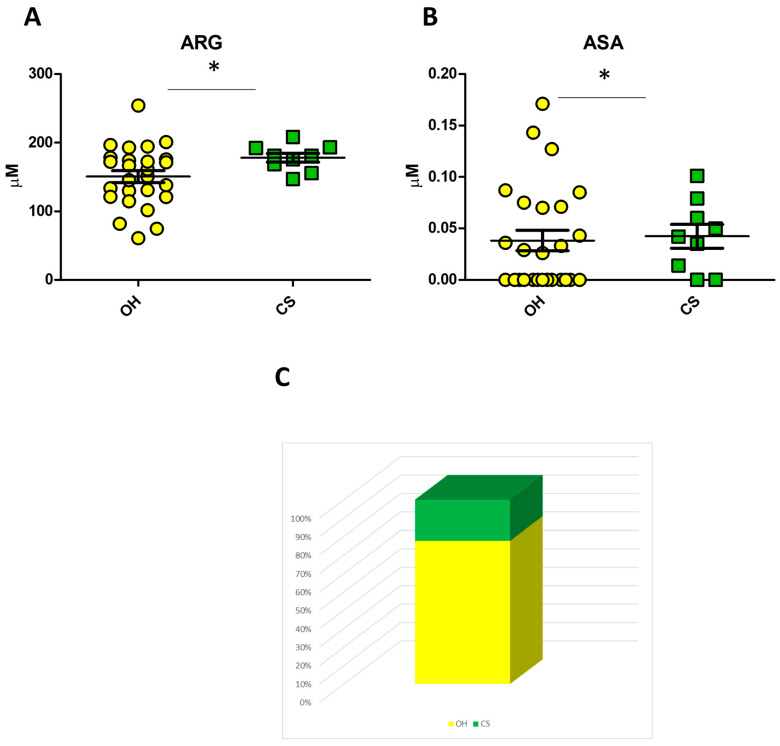
The logistic regression model with ARG and ASA-Total. Dot-plot showing plasma levels of ARG (**A**) and Asa-Total (**B**) in the comparison between the OH and CS groups; re-classification of US patients using ARG and ASA-Total simultaneously based on the application of the logistic regression algorithm described in Appendix A (**C**). * *p*-value < 0.05.

**Figure 5 biomedicines-12-02641-f005:**
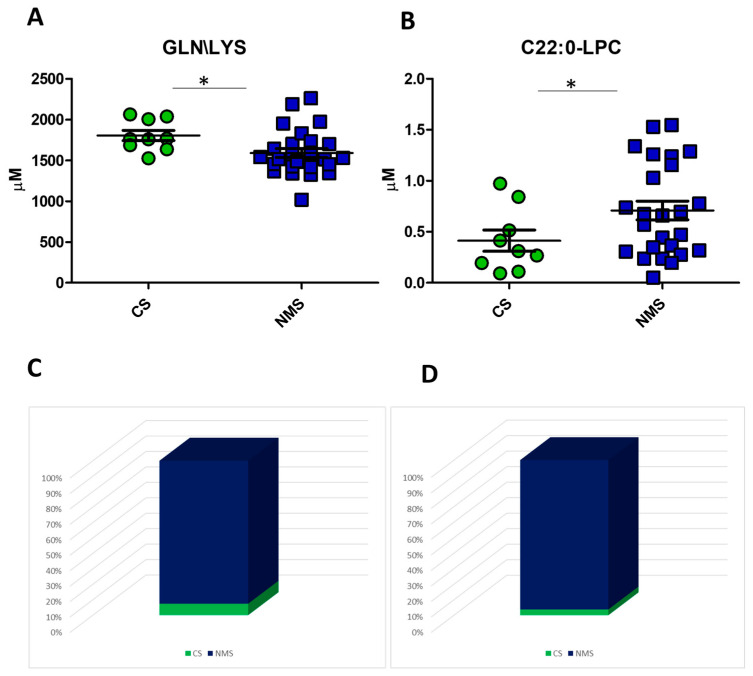
The logistic regression models with GLN/LYS and C22:0-LPC. Dot-plot showing plasma levels of GLN/LYS (**A**) and C22:0-LPC (**B**) in the comparison between the CS and NMS groups; re-classification of US patients using GLN/LYS plasma based on the application of the logistic regression algorithm described in Appendix A (**C**); re-classification of US patients using C22:0-LPC plasma based on applying the logistic regression algorithm described in Appendix A (**D**). * *p*-value < 0.05.

**Table 1 biomedicines-12-02641-t001:** Descriptive analysis.

Variables	CTRL Group (*n* = 10)	TLC Group				*p* Value
		CS (*n* = 9)	OH (*n* = 23)	NMS (*n* = 26)	US (*n* = 27)	
Age	64.5 ± 6.79	72 ± 16.0	60 ± 16.2	53.5 ± 13.7	58 ± 14.5	0.051
Sex (M/F)	4 (4.2%)/6 (6.3%)	5 (5.3%)/4 (4.2%)	11 (11.6%)/12 (12.6%)	12 (12.6%)/14 (14.7%)	13 (13.7%)/14 (14.7%)	0.975
Alcohol (n/y)	5 (5.35)/5 (5.3%)	6 (6.4%)/3 (3.2%)	18 (19.1%)/5 (5.3%)	21 (22.3%)/4 (4.3%9	19 (20.2%)/8 (8.5%)	0.302
Smoke (n/y)	9 (9.5%)/1 (1.1%)	4 (4.2%)/5 (5.3%)	20 (21.1%)/3 (3.2%)	15 (15.8%)/11 (11.6%)	14 (14.7%)/13 (13.75)	0.017
Exercise (n/y)	6 (6.5%)/4 (4.3%)	4 (4.3%)/5 (5.4%)	15 (16.1%)/8 (8.6%)	16 (17.2%)/9 (9.7%)	20 (21.5%)/6 (6.5%)	0.489
BH (n/y)	5 (5.3%)/5 (5.3%)	5 (5.3%)/4 (4.2%)	13 (13.7%)/10 (10.5%)	21 (22.1%)/5 (5.3%)	13 (13.7%)/14 (14.7%)	0.143
T2D (n/y)	10 (10.5%)/0 (0.0%)	7 (7.4%)/2 (2.1%)	20 (21.1%)/3 (3.2%)	25 (26.3%)/1 (1.1%)	22 (23.2%)/5 (5.3%)	0.273
CVD (n/y)	8 (8.4%)/2 (2.1%)	7 (7.4%)/2 (2.1%)	20 (21.1%)/3 (3.2%)	24 (25.3%)/2 (2.1%)	22 (23.2%)/5 (5.3%)	0.734
Weight (kg)	164 ± 8.48	164 ± 8.48	164 ± 8.48	164 ± 8.48	164 ± 8.48	0.179
Height (cm)	164 ± 8.48	168 ± 9.33	165 ± 35.4	170 ± 9.70	167 ± 9.26	0.688
BMI (kg/m^2^)	24.0 ± 4.18	24.6 ± 2.88	25.6 ± 3.49	24.1 ± 4.58	26.7 ± 4.27	0.072
SBP (mmHg)	125 ± 15.1	145 ± 12.1	134 ± 14.0	125 ± 13.5	129 ± 15.7	0.005
DBP (mmHg)	75.0 ± 6.67	81.9 ± 10.5	77.9 ± 9.36	78.6 ± 8.68	77.2 ± 10.8	0.529
waist (cm)	88.5 ± 11.5	91.5 ± 8.00	95 ± 11.7	87.0 ± 16.3	98.0 ± 12.3	0.141
IMT (μm)	825 ± 253	676 ± 128	597 ± 113	555 ± 134	648 ± 138	<0.001
Hb (g/dL)	13.3 ± 1.33	14.2 ± 1.51	14.2 ± 2.19	14.1 ± 1.60	14.0 ± 1.02	0.465
WBC (×10^9^/L)	5.70 ± 1.33	6.37 ± 2.35	6.06 ± 1.61	6.11 ± 2.18	6.95 ± 1.77	0.294
glyc (mg/dL)	90.5 ± 6.55	91 ± 23.4	97.5 ± 13.2	88.0 ± 11.2	94.5 ± 16.6	0.271
TRG (mg/dL)	81.5 ± 38.1	89.5 ± 43.6	88 ± 53.7	76.0 ± 47.2	102 ± 65.2	0.576
eGFR (mL/min)	83.5 ± 45.6	79.1 ± 18.4	89.8 ± 15.3	85.6 ± 19.3	79.5 ± 8.6	0.497
Tot-chol (mg/dL)	190 ± 37.8	192 ± 32.1	193 ± 34.0	195 ± 33.3	201 ± 41.7	0.618
HDL-chol (mg/dL)	67.5 ± 15.1	47.0 ± 21.0	50.0 ± 12.3	53.5 ± 13.1	52.5 ± 20.5	0.120
LDL-chol (mg/dL)	106 ± 32.8	116 ± 22.6	126 ± 31.7	124 ± 37.2	127 ± 44.0	0.387

Abbreviations: CTRL: control group; TLC: transient loss of consciousness; CS: cardiological syncope; OH: orthostatic syncope; NMS: neuromediated syncope; US: unexplained syncope; M/F: male/female; n/y: no/yes; BH: bold hypertension; T2D: type 2 diabetes; CVD: cardiovascular disease; BMI: body mass index; SBP: systolic blood pressure; DBP: diastolic blood pressure; IMT: intima-media thickness; Hb: hemoglobin; WBC: white blood cells; glyc: glycemia; TRG: triglycerides; eGFR: glomerular filtration rate; Tot-chol: total cholesterol; HDL-chol: HDL cholesterol; LDL-chol: LDL cholesterol.

**Table 2 biomedicines-12-02641-t002:** Pearson’s correlation between plasma metabolic profile and clinical features (SBP and IMT).

	SBP Pearson Correlation		c-IMT Pearson Correlation	
	r	R Squared	r	R Squared
ALA (µM)	0.143	0.020	−0.021	0.000
ARG (µM)	0.084	0.007	−0.069	0.005
CIT (µM)	0.108	0.012	0.034	0.001
GLN/LYS (µM)	0.064	0.004	0.073	0.005
GLU (µM)	0.012	0.000	−0.180	0.032
GLY (µM)	−0.126	0.016	−0.021	0.000
LEU\ILE\PRO-OH (µM)	0.006	0.000	−0.125	0.016
MET (µM)	−0.084	0.007	−0.163	0.026
METILDOPA (µM)	0.032	0.001	−0.070	0.005
ORN (µM)	−0.069	0.005	−0.030	0.001
PHE (µM)	−0.140	0.020	−0.191	0.036
PRO (µM)	0.041	0.002	−0.015	0.000
SA (µM)	−0.060	0.004	−0.070	0.005
TYR (µM)	0.046	0.002	−0.025	0.001
VAL (µM)	0.016	0.000	−0.125	0.016
ASA-Total (µM)	−0.087	0.008	−0.023	0.001
ADO (µM)	0.143	0.020	−0.005	0.000
C0 (µM)	0.125	0.016	0.279	0.078
C10 (µM)	0.050	0.003	0.107	0.011
C10:1 (µM)	0.074	0.006	0.057	0.003
C10:2 (µM)	0.087	0.008	−0.023	0.001
C2 (µM)	0.063	0.004	0.185	0.034
C3 (µM)	0.071	0.005	0.190	0.036
C3DC\C4OH (µM)	0.058	0.003	0.160	0.026
C4 (µM)	0.025	0.001	0.051	0.003
C4DC\C5OH (µM)	0.170	0.029	0.113	0.013
C5 (µM)	−0.108	0.012	0.053	0.003
C5:1 (µM)	0.111	0.012	0.010	0.000
C5DC\C6OH (µM)	−0.065	0.004	0.238	0.057
C6 (µM)	0.016	0.000	0.111	0.012
C6DC (µM)	0.090	0.008	0.314	0.099
C8 (µM)	0.038	0.001	0.138	0.019
C8:1 (µM)	−0.101	0.010	−0.076	0.006
D-ADO (µM)	−0.151	0.023	−0.080	0.006
C12 (µM)	0.053	0.003	0.077	0.006
C12:1 (µM)	0.156	0.024	0.133	0.018
C14 (µM)	0.058	0.003	0.094	0.009
C14:1 (µM)	0.047	0.002	0.098	0.010
C14:2 (µM)	0.048	0.002	0.053	0.003
C14OH (µM)	0.118	0.014	0.270	0.073
C16 (µM)	0.081	0.007	0.125	0.016
C16:1 (µM)	0.018	0.000	0.071	0.005
C16:1OH\C17 (µM)	0.053	0.003	0.232	0.054
C16OH (µM)	0.159	0.025	0.152	0.023
C18 (µM)	0.094	0.009	0.031	0.001
C18:1 (µM)	0.068	0.005	0.071	0.005
C18:1OH (µM)	0.102	0.011	0.158	0.025
C18:2 (µM)	−0.002	0.000	−0.039	0.002
C18:2OH (µM)	0.091	0.008	0.096	0.009
C18OH (µM)	0.116	0.014	0.080	0.006
C20 (µM)	−0.054	0.003	0.105	0.011
C20:0-LPC (µM)	−0.032	0.001	−0.084	0.007
C22 (µM)	0.047	0.002	0.008	0.000
C22:0-LPC (µM)	−0.030	0.001	−0.050	0.002
C24 (µM)	−0.147	0.022	−0.049	0.002
C24:0-LPC (µM)	−0.054	0.003	−0.107	0.011
C26 (µM)	−0.030	0.001	−0.070	0.005
C26:0-LPC (µM)	0.009	0.000	0.023	0.001
CIT/ARG	−0.054	0.003	0.043	0.002
TYR/CIT	−0.049	0.002	−0.048	0.002
PHE/TYR	−0.143	0.020	−0.163	0.027
C0/(C16 + C18)	0.040	0.002	0.132	0.018
C14:1/C16	0.072	0.005	0.089	0.008
C14:1/C2	0.014	0.000	0.052	0.003
C16OH/C16	0.075	0.006	0.065	0.004
C18OH/C18	0.058	0.003	0.116	0.014
C5OH/C10	−0.149	0.022	0.034	0.001
C5OH/C2	−0.107	0.011	−0.106	0.011
C5OH/C8	−0.160	0.026	0.014	0.000
C5/C0	−0.228	0.052	−0.107	0.011
C5/C2	−0.178	0.032	−0.068	0.005
C5/C3	−0.168	0.028	−0.085	0.007
C5DC/C16	−0.132	0.017	0.085	0.007
C5DC/C8	−0.247	0.061	0.060	0.004
LEU/ALA	−0.163	0.027	−0.149	0.022
LEU/PHE	0.110	0.012	0.175	0.031
MET/PHE	−0.024	0.001	0.188	0.035
VAL/PHE	0.115	0.013	0.217	0.047
C3/C2	−0.063	0.004	−0.021	0.000
C3/C16	0.008	0.000	0.063	0.004
C3/MET	0.144	0.021	0.282	0.079
C8/C2	0.066	0.004	0.035	0.001
C8/C10	−0.007	0.000	0.119	0.014
(C16 + C18.1)/C2	−0.140	0.020	−0.075	0.006

## Data Availability

The research material’s source documents are paper and electronic files, and all datasets are deposited at the Day Hospital Service of Tor Vergata University Hospital (Rome, Italy). They are available upon reasonable request to the corresponding author.

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
