# Peer review of "A Metabolomic Approach to Unexplained Syncope"

_biomedicines, 2024, doi:10.3390/biomedicines12112641_

Round 1
Reviewer 1 Report
Comments and Suggestions for Authors
The aim of this work is to evaluate the relationship between metabolic profile and syncope. Syncopes were divided into several groups, syncopes of cardiac origin (CS: n=9), neurohumoral syncopes (NMS n= 26, unexplained syncopes (US n=27), orthostatic hypotensions (OH n= 23). The authors measured plasma concentrations of amino acids (mainly Phe, Arg, Val, met, Leu, Ileu, Pro, ornithine, citruline and some derivatives) as well as acyl carnitines and LPCs. The results show that there is an increase in Phe, Arf and Orn in syncopes on pathological heart, an increase in Phe in the OH and US group, an increase in val, Leu and ILeu in the NMS and TLC group. Moreover, the concentrations of C22 LPC and the GLN/Lys ratio would allow to distinguish the US group from the CS group with satisfactory ROC curves.
The study is interesting but there are serious biases.
Major: regarding the classification, it is confusing/
For example TLC is a symptom not a disease. Moreover what difference do the authors make between unexplained syncope (US) and NMS?. Sometimes they used NMS sometimes TLC, please harmonize. There is no mention of the "metabolic" classification on the adenosine concentration which allows to differentiate NMS syncopes with low adenosine, those with high adenosine and situational syncopes which are pure cholinergic dependent syncope (See Guieu R et al JACC J Am Coll Cardiol. 2015 Jul 14;66(2):204-5. doi: 10.1016/j.jacc.2015.04.066).
This is important because adenosine comes partly from the methionine cycle and there could be a link between these groups of syncopes.
The control group is low n = 10 as well as the CS group (n = 9) which does not allow to draw conclusions from the ROC curves, which should in fact be deleted.
The conclusion on the role of LPC is not acceptable because as highlighted in the discussion lines 320 to 325, LPC is involved in a large number of inflammatory, neurodegenerative pathologies etc. which makes it a marker of very low specificity.
Finally the metabolomic profile may result in other pathologies not detected in the different groups like HTA. As exemples there is a difference in SBP between Controls and CS or other patient groups. Can the author assume that HTA is not responsible for the different metabolomic profiles?
Minor
in table S1 why is the Met/Phe ratio used rather than the concentrations separately? No unit is given in this table I assume arbitrary units?
moreover the differences in the concentrations of amino acids are very small even if they are significant? Therefore the units should be given.
Table S2 the meaning of the abbreviations FDR, CS should be specified
Author Response
Comments 1: regarding the classification, it is confusing/For example TLC is a symptom not a disease. Moreover what difference do the authors make between unexplained syncope (US) and NMS?. Sometimes they used NMS sometimes TLC, please harmonize. There is no mention of the "metabolic" classification on the adenosine concentration which allows to differentiate NMS syncopes with low adenosine, those with high adenosine and situational syncopes which are pure cholinergic-dependent syncope (See Guieu R et al JACC J Am Coll Cardiol. 2015 Jul 14;66(2):204-5. doi: 10.1016/j.jacc.2015.04.066).
This is important because adenosine comes partly from the methionine cycle and there could be a link between these groups of syncopes.
Our response: We sincerely thank the Reviewer for this valuable comment. Your important and focused suggestion will let us substantially improve the overall quality of our manuscript. We improved the introduction by adding the sentences (pages 1-2, lines 36-68):
“Syncope is a transient loss of consciousness (TLC) characterized by rapid onset, short duration, and spontaneous reversibility with complete recovery. It is a symptomatic consequence of global cerebral hypoperfusion triggered by vasovagal stimulation, arterial hypotension, or low systolic output.”
Despite its high frequency, obtaining an accurate estimate of the incidence of syncope is difficult because of the different definitions used among studies and because most syncopal events do not require medical attention [1]. According to the guidelines of the European Society of Cardiology (ESC) of 2018 [2], syncope is classified into:
- reflex or neurologically mediated syncope (NMS), related to a specific trigger;
- syncope due to orthostatic hypotension (OH), defined as a drop >20 mmHg in systolic blood pressure (SBP) or >10 mmHg in diastolic blood pressure after standing for three minutes;
- cardiac syncope (CS), caused by arrhythmic pathologies or structural diseases of the heart and great vessels.
“NMS can be further classified into vasovagal syncope (emotional or due to prolonged standing), situational syncope (related to a specific trigger), syncope due to stimulation of the carotid sinus (carotid sinus syndrome), and non-classical forms that occur without the typical prodromes or triggers of NMS [2]. To clarify the pathophysiological mechanisms underlying these events and to reclassify some unexplained syncopal events as NMS, the evaluation of plasma adenosine levels and the characterization of the adenosine A2A receptor have been proposed [Guieu R, Deharo JC, Ruf J, Mottola G, Kipson N, Bruzzese L, Gerolami V, Franceschi F, Ungar A, Tomaino M, Iori M, Brignole M. Adenosine and Clinical Forms of Neurally-Mediated Syncope. J Am Coll Cardiol. 2015 Jul 14;66(2):204-5. doi: 10.1016/j.jacc.2015.04.066].Specific purinergic profiles characterize different forms of NMS that can be classified as low-, normal-, and high-adenosine syncope. This classification not only provides a deeper understanding of NMS but also offers hopeful therapeutic implications.
Syncopal episodes that are difficult to categorize according to the guidelines or the metabolic classifications are often found even after further diagnostic investigations in the Syncope Units. These episodes remain unexplained syncope (US) [3]. US represents a crucial challenge in identifying the risk of short-term adverse outcomes, as it is difficult to determine whether the event will follow a benign course, such as in NMS, or if it is at risk of major cardiovascular events.”
Comments 2: The control group is low n = 10 as well as the CS group (n = 9) which does not allow to draw conclusions from the ROC curves, which should in fact be deleted
Our response: We cordially thank the Reviewer for this notable comment because it allowed us to enrich our discussion by adding the following sentence: (page 21, lines 473-481)
“The same limitations have made it difficult to collect healthy patients to include in the CTRL groups and subjects with CS. Patients were enrolled in the Day Hospital Service, where patients with suspected CS but at low risk of short-term adverse outcomes were referred. Most patients with CS are at high risk of short-term adverse outcomes and are therefore admitted to a cardiology care setting. However, we included the CS group in our analysis to make our research more comprehensive. We believe that the findings of this pilot study will pave the way for further research with a larger sample, allowing us to validate the model used and potentially make significant contributions to the field.”
Comments 3: The conclusion on the role of LPC is not acceptable because as highlighted in the discussion lines 320 to 325, LPC is involved in a large number of inflammatory, neurodegenerative pathologies etc. which makes it a marker of very low specificity
Our response: We sincerely thank the Reviewer for this important comment because it allowed us to better explain our point of view. Although LPCs are implicated in several pathologies, strong evidence recognizes their role in neurodegenerative and cardiometabolic diseases. Furthermore, the statistical robustness of our observations confirms its diagnostic reliability as a biomarker, a finding with significant practical implications. We have better explained our belief by modifying the text as follows (page 19, lines 385-387):
“Although LPCs are involved in several pathological scenarios, they are increasingly recognized as a critical factor associated with neurodegenerative and cardiometabolic diseases due to their proinflammatory and pro-atherogenic actions [30]”
Furthermore, we have specified the limitations on page 22 and lines 492-494:
“Our findings strongly indicate that C22:0-LPC and GLN/LYS are highly effective tools for reclassifying US. Nonetheless, due to the limitations highlighted earlier, further evaluations are essential to fully confirm our pilot study's results.”
Comments 4: Finally the metabolomic profile may result in other pathologies not detected in the different groups like HTA. As exemples there is a difference in SBP between Controls and CS or other patient groups. Can the author assume that HTA is not responsible for the different metabolomic profiles?
Our response: We thank the Reviewer for this targeted comment because it allowed us to improve our manuscript. Although the SBP and c-IMT values ​​were significantly different among the five groups considered, we performed a Pearson correlation between plasma metabolites c-IMT and SBP, obtaining no moderate or strong correlation, as can be seen from the Pearson coefficient values ​​reported in the new Table 2 (pages 7-9 line 229).
Minor
Comments 5: in table S1 why is the Met/Phe ratio used rather than the concentrations separately? No unit is given in this table I assume arbitrary units? moreover the differences in the concentrations of amino acids are very small even if they are significant? Therefore the units should be given.
Our response: We would like to thank the Reviewer for this focused and precise comment. The analyses performed are related to a panel of metabolites composed of amino acids, free carnitine, acylcarnitines, ketones nucleosides, and lysophospholipids, as well as metabolic ratios. The method used is based on the quantification of these metabolites expressed in µM. We apologize for not having clarified these points sufficiently, and according to the Reviewer's comment, we have added Table S1 with the list and relative abbreviations of all quantified metabolites, as well as the MS/MS transitions used and the internal standards used for quantification. We have also listed all quantified values ​​in the entire case series in Table S2. In addition, we have added the units of measurement to Table 1 and Table S3 (Table S1 in the first version of the manuscript).
Comments 6: Table S2 the meaning of the abbreviations FDR, CS should be specified.
Our response: We would like to thank the Reviewer for this targeted comment.
We carefully reviewed the text and corrected the mistakes in abbreviations.
Reviewer 2 Report
Comments and Suggestions for Authors
I will reccomend this important key submission to be accepted after adreesing the following minor comments
1, Line14, neurologically mediated syncope (NMS) should be noted together when it is used in text for the first time;
2, Line 47, word "nearly" could be changed to "over", becasue 36.6% is larger than 1/3?
3, Line 82, only one paper 25 seems not enough for your foot standing of this work in terms of medicine and clinical practice ? more citations are advised if possible ?And your new focus related to US as metabolomic tool should be refered here with more description as scientific clue or idea? showing us more thinking via medicne and clinical logistically?
4, Line119-120, blood drawing or taking should be on an empty stomach is necessary or not, pls note here just with a phrase.
5, Line 124, ... centrifugation for serum sampling..., some parameters as rpm or temperture should be noted and including necessary reference if possible?
6, Line 176-177, ... smoking (p=0.017). They also differed in SBP measured at the first visit (p=0.005) and c-IMT (p 0.001). ..., by traditioanl statistical tool, p value is used as threshold by less than as < or scope instead of equal symbol as =? plc check and confirm? and other similar situation or errors should be checked through in your whole text , as in line 200, 206 and 208, and other sections?
7, Figure 1, it shoukld be enlarged in size or ratio to be more clearly and seen easily? So do the Fgure 2?
8, As a whole of section Results, some more quanititative analysis should be enhanced focusing you five Figures if possible?
9, Any or more tables could be suggested except Table 1 if posible, it can present more information different from Figures perhaps? Or some key ones of 9 supplementary Tabels copuld be moved into main text if poissible and necessary?
10, Be sure all full and exact author information should be placed on the first page of supplementary files, and all should be cited in your main text.
11, Finally, may I ask a another question from my personal meaning, Could mics analysis be run within very short time length or very quick to meet the least or quickest requirement for UN diagnosis during ED period in clinical practice? Or any better option or choice to improve next you can give?
(END at 16 Oct 2024)
Author Response
Comments 1: Line14, neurologically mediated syncope (NMS) should be noted together when it is used in text for the first time;
Our response: We would like to thank the Reviewer for this targeted comment.
We carefully reviewed the text and corrected the mistakes in abbreviations.
Comments 2: Line 47, word "nearly" could be changed to "over", becasue 36.6% is larger than 1/3?
Our response: We would like to thank the Reviewer for this targeted comment. It is an important consideration and we changed the text on the page 2, line 73
Comments 3: Line 82, only one paper 25 seems not enough for your foot standing of this work in terms of medicine and clinical practice ? more citations are advised if possible ?And your new focus related to US as metabolomic tool should be refered here with more description as scientific clue or idea? showing us more thinking via medicne and clinical logistically?
Our response: We sincerely thank the Reviewer for this valuable comment. We added other citations to support our background considerations. Page 3, line 109-110
- 27 Savani G, Singh V, Rodriguez A, Tegene T, Cohen M, Alfonso C, Mitrani R, Viles-Gonzalez J, Myerburg R, Goldberger J. MORTALITY RATES AND COST OF HOSPITAL ADMISSIONS FOR SYNCOPE IN THE UNITED STATES. JACC. 2017 Mar, 69 (11_Supplement) 524. https://doi.org/10.1016/S0735-1097(17)33913-X
- 28 Probst MA, Su E, Weiss RE, Yagapen AN, Malveau SE, Adler DH, Bastani A, Baugh CW, Caterino JM, Clark CL, Diercks DB, Hollander JE, Nicks BA, Nishijima DK, Shah MN, Stiffler KA, Storrow AB, Wilber ST, Sun BC. Clinical Benefit of Hospitalization for Older Adults With Unexplained Syncope: A Propensity-Matched Analysis. Ann Emerg Med. 2019 Aug;74(2):260-269. doi: 10.1016/j.annemergmed.2019.03.031
- 29 Joy PS, Kumar G, Olshansky B. Syncope: Outcomes and Conditions Associated with Hospitalization. Am J Med. 2017 Jun;130(6):699-706.e6. doi: 10.1016/j.amjmed.2016.12.030
- 30 Firouzbakht T, Shen ML, Groppelli A, Brignole M, Shen WK. Step-by-step guide to creating the best syncope units: From combined United States and European experiences. Auton Neurosci. 2022 May;239:102950. doi: 10.1016/j.autneu.2022.102950.
- 31 Tannenbaum L, Keim SM, April MD, Long B, Koyfman A, Mattu A. Can I Send This Syncope Patient Home From the Emergency Department? J Emerg Med. 2021 Dec;61(6):801-809. doi: 10.1016/j.jemermed.2021.07.060.
- 32 Shen WK, Sheldon RS, Benditt DG, Cohen MI, Forman DE, Goldberger ZD, Grubb BP, Hamdan MH, Krahn AD, Link MS, Olshansky B, Raj SR, Sandhu RK, Sorajja D, Sun BC, Yancy CW. 2017 ACC/AHA/HRS Guideline for the Evaluation and Management of Patients With Syncope: A Report of the American College of Cardiology/American Heart Association Task Force on Clinical Practice Guidelines and the Heart Rhythm Society. Circulation. 2017 Aug 1;136(5):e60-e122. doi: 10.1161/CIR.0000000000000499. Epub 2017 Mar 9. Erratum in: Circulation. 2017 Oct 17;136(16):e271-e272. doi: 10.1161/CIR.0000000000000537
- 33 Li J, Smyth SS, Clouser JM, McMullen CA, Gupta V, Williams MV. Planning Implementation Success of Syncope Clinical Practice Guidelines in the Emergency Department Using CFIR Framework. Medicina (Kaunas). 2021 Jun 3;57(6):570. doi: 10.3390/medicina57060570.
Furthermore, to better clarify our purpose, we have also added to the text (pages 3, lines 111-112):
“In our study, we applied an omics technique to deepen our knowledge of the true nature of the US and suggest a practical application in hospital management.”
Comments 4: Line119-120, blood drawing or taking should be on an empty stomach is necessary or not, pls note here just with a phrase.
Our response: We would like to thank the Reviewer for this targeted comment. We have specified the withdrawal method on page 4, lines 149-150
“Blood samples were collected after an overnight fast to prevent test result interference.”
Comments 5: Line 124, ... centrifugation for serum sampling..., some parameters as rpm or temperture should be noted and including necessary reference if possible?
Our response: We thank the Reviewer for pointing out this oversight. We added this information on page 4 lines 155-156:
“About 8 ml of whole blood underwent centrifugation for serum sampling for 40 min at 4°C at 1000 rcf.”
Comments 6: Line 176-177, ... smoking (p=0.017). They also differed in SBP measured at the first visit (p=0.005) and c-IMT (p 0.001). ..., by traditioanl statistical tool, p value is used as threshold by less than as < or scope instead of equal symbol as =? plc check and confirm? and other similar situation or errors should be checked through in your whole text , as in line 200, 206 and 208, and other sections?
Our response: We would like to thank the reviewer for this targeted comment. We carefully reviewed the text and corrected the mistakes on pages 6-18 lines 213-350
Comments 7: Figure 1, it shoukld be enlarged in size or ratio to be more clearly and seen easily? So do the Fgure 2?
Our response: Following the Reviewer observation, we have modified figures 1 and 2 to make them more easily readable
Comments 8: As a whole of section Results, some more quanititative analysis should be enhanced focusing you five Figures if possible?
Our response: We thank the Reviewer for this comment. As reported in our previous work regarding the analytical evaluation of our method (Metabolites. 2021 Jul 22;11(8):473. doi: 10.3390/metabo11080473.), in terms of repeatability and precision, we cannot express the quantitative results obtained with more than three “figures”.
Comments 9: Any or more tables could be suggested except Table 1 if posible, it can present more information different from Figures perhaps? Or some key ones of 9 supplementary Tabels copuld be moved into main text if poissible and necessary?
Our response: We would like to thank the Reviewer for this targeted comment. We have added Table 2 to show Pearson's correlation between plasma metabolic profile and clinical characteristics (SBP and c-IMT). In the supplementary materials, we have also added Table S1 with the list and relative abbreviations of all quantified metabolites, as well as the MS/MS transitions used and the internal standards used for quantification.
Comments 10: Be sure all full and exact author information should be placed on the first page of supplementary files, and all should be cited in your main text.
Our response: We would like to thank the Reviewer for this precise comment. We reviewed the text and ensured that all complete and accurate information about the authors was included on the first page of the "supplementary materials" files and that all tables and figures were cited in the main text.
Comments 11: Finally, may I ask a another question from my personal meaning, Could mics analysis be run within very short time length or very quick to meet the least or quickest requirement for UN diagnosis during ED period in clinical practice? Or any better option or choice to improve next you can give?
Our response: We would like to thank the Reviewer for giving us the opportunity to further explore this point. In fact, the method we presented is already in clinical use for the evaluation of inherited metabolic diseases in the context of the expanded newborn screening program, as reported in several works (10.1016/j.clinbiochem.2005.12.009, doi: 10.3390/metabo10020044, doi: 10.3390/ijerph17103601). The analysis times are really very fast considering that the analysis of the metabolic profile is performed in less than two minutes, also considering the sample preparation, so in less than two hours the laboratory is able to provide the patient's metabolic profile. This represents a huge strength considering its application in the clinical field to assist physicians in a rapid and precise diagnosis especially in the perspective of precision medicine. We have included this consideration in the discussion of this manuscript (page 21, lines 460-468).